# Testing and Prescribing Vitamin B12 in Swiss General Practice: A Survey among Physicians

**DOI:** 10.3390/nu13082610

**Published:** 2021-07-29

**Authors:** Katarina Bardheci, Levy Jäger, Lorenz Risch, Thomas Rosemann, Jakob M. Burgstaller, Stefan Markun

**Affiliations:** 1Institute of Primary Care, University of Zurich, University Hospital Zurich, 8091 Zurich, Switzerland; Katarina.Bardheci@usz.ch (K.B.); Levy.Jaeger@usz.ch (L.J.); thomas.rosemann@usz.ch (T.R.); jakob.burgstaller@usz.ch (J.M.B.); 2Labormedizinisches Zentrum Dr Risch AG, 9470 Buchs, Switzerland; lorenz.risch@risch.ch

**Keywords:** vitamin B12, cobalamin, survey, general practitioners, laboratory testing, polyneuropathy, anemia, idiopathic fatigue, cognitive, depression

## Abstract

Testing and prescribing vitamin B12 (also known as cobalamin) is increasing in Switzerland but substantial variation among general practitioners (GPs) with respect to testing has been noted. In this study, we aimed at exploring GPs’ mindsets regarding vitamin B12 testing and prescribing. A cross-sectional study was conducted using an online survey distributed by e-mail to Swiss GPs. The questionnaire explored mindsets related to testing and prescribing vitamin B12 in specific clinical situations, as well as testing and prescribing strategies. The questionnaire was sent to 876 GPs and 390 GPs responded (44.5%). The most controversial domains for testing and prescribing vitamin B12 were idiopathic fatigue (57.4% and 43.4% of GPs agreed, respectively) and depressive symptoms (53.0% and 35.4% of GPs agreed, respectively). There was substantial variation among GPs with regard to testing strategies (89.5% of GPS used a serum cobalamin test, 71.3% of GPS used holotranscobalamin, and 27.6% of GPs used homocysteine or methylmalonic acid). Intramuscular injection was the predominantly prescribed route of application (median of 87.5% of the prescriptions). In this study, we focus on discordant mindsets that can be specifically targeted by using educational interventions, and research questions that still need answering specifically about the effectiveness of vitamin B12 for idiopathic fatigue.

## 1. Introduction

Vitamin B12 (also known as cobalamin) is an essential cofactor for metabolic processes such as DNA synthesis or methylation and is indispensable for bodily functions such as blood formation and for the nervous system [1]. Clinical manifestations of vitamin B12 deficiency include anemia and peripheral neuropathy and, in advanced cases, permanent and severe complications can occur such as subacute combined degeneration [2]. However, determination of vitamin B12 deficiency is difficult, especially in early stages, because non-specific symptoms such as fatigue may appear or even no symptoms at all, i.e., a state referred to as subclinical vitamin B12 deficiency [3]. Further contributing to diagnostic uncertainties is the discriminatory power of existing laboratory tests for vitamin B12 deficiency. Serum cobalamin assays alone do not discriminate between biologically active (holotranscobalamin) and inactive (holohaptocorrin) protein bound forms of vitamin B12, and therefore are of limited value [4]. A serum cobalamin cut-off point at 148 pmol/L is widely accepted due to its high specificity for vitamin B12 deficiency, however, sensitivity of this cut-off is low and a borderline area of serum cobalamin ranging up to 350 pmol/L must be considered wherein deficiency cannot be ruled out with sufficient certainty [5]. Strategies aimed at measurements of holotranscobalamin and secondary metabolic products accumulating in vitamin B12 deficiency (methylmalonic acid or homocysteine) have been proposed to more accurately identify vitamin B12 deficiency [6]. These measures, however, are themselves influenced by ulterior factors such as prevalent comorbidities [7], other vitamin deficiencies [1], or analytical interference [8], therefore, decreasing their specificity.

Due to modest diagnostic accuracy of laboratory tests and a multitude of potential but non-specific clinical signs, vitamin B12 deficiency often cannot be ruled out with certainty. Especially general practitioners (GPs) are often confronted with non-specific clinical signs such as fatigue, depressive symptoms, or cognitive decline and how they account for potential vitamin B12 in these situations is unknown [9,10,11]. Interestingly, however, serum cobalamin is among the laboratory tests with the highest between-physician variation in Swiss general practice [12] and serum cobalamin tests are increasing in Switzerland, with almost 20% of all Swiss adult citizens tested in 2018 [13]. Knowledge about existing mindsets triggering testing and prescribing vitamin B12 is required to identify whether specific misconceptions or lack of knowledge contribute to this variation and increase. Therefore, the aim of this study was to better understand GPs’ mindsets regarding vitamin B12 testing and prescribing.

## 2. Materials and Methods

### 2.1. Design, Setting, and Participants

We conducted a cross-sectional study surveying Swiss GPs. Potential participants were members of a convenience sample of GPs affiliated with the Institute of Primary Care at the University of Zurich. The sample consisted of (1) GPs who were members of the FIRE project [14] (n = 480, GPs who were from the FIRE project practice in different parts of Switzerland and contributed routine data from their electronic medical records for research purposes); (2) respondents of a previous outreach among all GPs in the Canton of Zurich who answered to be generally interested in participating in surveys (n = 96); (3) GPs involved in undergraduate and postgraduate medical training organized by the Institute of Primary Care (n = 300). Survey participation was incentivized by a draw for one of three 500 CHF prizes. According to Swiss federal law, approval of the Ethical Committee is not necessary for surveys among physicians [15].

### 2.2. Instruments and Procedures

We conducted the survey using the survey platform Surveymonkey^®^ (Momentive Inc., San Mateo, CA, USA). We reached out to the physicians in the sample by sending out an e-mail invitation to the group on three different dates. The first invitation was sent on 26 November 2020, further invitations were sent one week and six weeks after the first invitation. Each invitation e-mail contained a participation button which directly linked to the online survey platform and entry of a personal code for each GP was used to ensure completion by respective individuals only and to prevent multiple completion. The survey was available for completion for a total of eight weeks and was closed on 22 January 2021.

The questionnaire was in the German language, self-administered, and consisted of 14 mandatory questions (4 multiple-choice, 4 single-choice, 1 Likert-type, and 5 sliding scale questions). Questions were organized into three parts and their sequence was not randomized. The first part (Questions 1–4) asked for GPs’ mindsets regarding the role of vitamin B12 testing and prescribing in specific clinical situations (i.e., suspected peripheral polyneuropathy, anemia, idiopathic fatigue, cognitive complaints, depressive symptoms, and check-up of asymptomatic patients, in Questions 3 and 4 additionally, hair loss, oral aphthae, and the known risk for malnutrition/malabsorption). The second part (Questions 5–9) asked for GPs’ testing strategies concerning vitamin B12 (first- and second-line parameters used for sufficiency assessment of bodily vitamin B12 storages), patient involvement in decisions to test or prescribe vitamin B12 (proportions solely based on patient request versus physician recommendation), prescribed routes of application (proportion of enteral versus parenteral application), and confidence in effectiveness of vitamin B12 prescriptions (proportions prescribed as a placebo versus no placebo). The third part (Questions 10–14) asked demographic information. At the end of the questionnaire, further optional questions were added concerning future research collaboration and to allow participation in the draw by including the necessary contact information. These additional questions were outside the scope of this study and not reported in this study (Appendix A for the original questionnaire and Appendix A for the English translation). The questionnaire was pilot tested by 9 persons representing the target population to enhance understandability and their feedback was used for improvement of the survey.

### 2.3. Statistics

We explored GPs’ mindsets regarding the role of vitamin B12 testing and prescribing in specific clinical situations using the first part of the survey. For mindsets involving testing alone, we calculated a composite score for testing willingness using responses to the Likert-type question and assigning numerical values to agreement ratings for testing in each clinical situation (ranging from 0, “fully disagree”, to 4, “fully agree”, thus, giving the score a possible range from 0 to 24). Principal component analysis was used to assess the dimensionality of scores using scree testing (Appendix A) [16]. Internal consistency was quantified by means of Cronbach’s alpha, with a value above 0.7 considered to be satisfactory [17,18]. We modeled the score by means of linear regression with adjustment for gender, experience as a GP in years, practice of complementary and alternative medicine (CAM), practice of psychosomatic medicine, and indicator variables for values <50% in the responses to questions 6 and 8. Results were reported in terms of coefficient estimates and corresponding 95% confidence intervals (CI). Statistical significance was assessed by means of *p*-values using a threshold of 0.05. Further items of the questionnaire were summarized using descriptive statistics and graphical representation. Categorical data were summarized as counts (n) and proportions (%), numerical data as median and interquartile range (IQR). Missing data were left unchanged and reported if exceeding 5%. All analyses were conducted using R 4.0.3 (R Foundation for Statistical Computing, Vienna, Austria), the packages likert, ggplot2, and networkD3 were used for figure editing, the package psy for scale analysis [19,20,21,22,23].

## 3. Results

### 3.1. Sampling and Sample Characteristics

The survey was distributed to 876 GPs and 390 of them responded (response rate of 44.5%). Among the responding GPs, 376 GPs participated (participation rate 96.4%) and among all participants, 347 GPs completed the survey (92.3% completion rate). Survey participants were 40.7% female and had a median working experience of 23.0 years (IQR 16.0 to 30.0 years). A majority of 306 (81.4%) participants worked in a group practice and 345 (91.8%) of the participants had board certification in general internal medicine. Further characteristics of the participants are listed in Table 1.

### 3.2. GPs’ Mindsets Regarding the Role of Vitamin B12 Testing and Treatment in Specific Clinical Situations

Agreement with testing of vitamin B12 was highest in suspected peripheral polyneuropathy (99.2% of participants rather or fully agreed), cognitive symptoms (87.4%), and anemia (85.4%). Slightly more than half of GPs agreed with testing of vitamin B12 in idiopathic fatigue (57.4%) and depressive symptoms (53.0%). There was little agreement among GPS for testing vitamin B12 as part of a check-up in asymptomatic patients (12.4%). See Figure 1 for detailed results of agreement with testing vitamin B12 in different clinical situations.

The median score for vitamin B12 testing willingness was 17 (IQR 14–20). Internal consistency was considered to be satisfactory with Cronbach’s alpha equal to 0.71. Multivariable linear regression showed significantly lower scores for female respondents (coefficient estimate 1.02, 95% CI 0.15 to 1.90), low patient involvement in decisions to test vitamin B12 (coefficient estimate for <50% of decisions to test solely based on patient request, 1.48, 95% CI 0.40–2.56) and high confidence in effectiveness (coefficient estimate for <50% of prescriptions being placebo interventions, 1.78, 95% CI 0.31–3.25). No significant association was found for years of experience as a GP, patient involvement in decisions to vitamin B12 prescription, and practice of CAM or psychosomatic medicine (Appendix A).

Agreement with prescribing vitamin B12 when test results are within a borderline area was highest for suspected peripheral polyneuropathy (81.1% of participants rather or fully agreed), for cognitive symptoms (65.7%), and for anemia (64.4%). Slightly less than half of GPs agreed with such prescribing for idiopathic fatigue (43.4%), and depressive symptoms (35.4%). Less than 10% of GPs agreed with such prescribing for other clinical situations (malnutrition/malabsorption 5.1%, after check-up in asymptomatic patients 3.7%, hair loss/oral aphthae 0.3%, and other 8.8%). Among the GPs, 10.6% of the GPs answered that they categorized borderline test results as normal results.

Concerning the three most common clinical situations where the vitamin B12 status was actually measured in the past year, most frequently, participants mentioned investigation of suspected peripheral polyneuropathy (77.1%), known risk for malnutrition/malabsorption (60.1%), and anemia (49.7%). The frequencies of clinical situations wherein vitamin B12 was prescribed were similar to the ones where it was measured (see Table 2 for the most common clinical situations for testing and prescribing vitamin B12)**.**

### 3.3. Testing Strategy, Route of Application, Patient Involvement, and Confidence in Effectiveness

Most often GPs reported using serum cobalamin (89.8%) and holotranscobalamin (71.3%) for measuring vitamin B12, and more rarely homocysteine or methylmalonic acid (27.6%). With respect to testing strategies, the majority of GPs reported following a multi-step approach (72.9%). Among specific strategies, the three most common were: first-line assessment with serum cobalamin and second-line assessment with holotranscobalamin (37.8%), serum cobalamin only (15.7%) assessment, and first-line assessment with serum cobalamin and second-line assessment with holotranscobalamin and homocysteine or methylmalonic acid (14.1%). Figure 2 illustrates testing strategies reported by GPs.

Concerning the route of application, GPs responded to prescribe parenteral vitamin B12 (e.g., as intramuscular injection) in a median of 87.5% (IQR 65–95%) of the cases. Regarding patient involvement, GPs responds indicated that a median of 20% (IQR 10–40%) of the decisions to test vitamin B12 was solely based on a patient’s request and a median of 10% (IQR 0–20%) of the decisions to prescribe vitamin B12 was solely based on a patient’s request. Regarding confidence in effectiveness, GPs responds indicated that a median of 5% (IQR 0–20%) of the vitamin B12 prescriptions were intended as placebo interventions.

## 4. Discussion

Vitamin B12 testing and prescribing are increasing in Switzerland with considerable variation among GPs. In this cross-sectional study, we aimed to assess mindsets behind testing and prescribing vitamin B12 in Swiss GPs. Most GPs agreed with the notion that suspected peripheral polyneuropathy always requires vitamin B12 testing, and consensually, this clinical situation was the most frequently reported reason for testing and prescribing vitamin B12. Approximately half of the GPs supported testing vitamin B12 in cases with idiopathic fatigue and depressive symptoms as well as prescribing vitamin B12 in cases with idiopathic fatigue and borderline vitamin B12 test results. Testing strategies for vitamin B12 varied significantly among GPs and among routes of application, mostly intramuscular injections were used. Initiatives to test and prescribe vitamin B12 seemed to be, in most cases, driven by GPs rather than patients and, most often, GPs expected specific effects rather than placebo effects when prescribing vitamin B12.

The usefulness of tests in specific clinical situations depends on the pretest probability to confirm or rule out specific conditions and the availability of effective treatments that can be applied depending on test outcomes. Regarding vitamin B12, pretest probabilities for vitamin B12 deficiency vary significantly among specific clinical situations. Although vitamin B12 deficiency can rarely be identified to be the definitive cause of peripheral polyneuropathy [24], cognitive decline [25] or anemia [26], testing can reveal these cases and prompt effective treatment by replacing vitamin B12 [27]. In our survey and in concordance with guidelines, on the one hand, the vast majority of GPs agreed with testing vitamin B12 in the three above-mentioned clinical situations. On the other hand, however, half of GPs agreed with testing of vitamin B12 in idiopathic fatigue, which, in contrast, is discouraged by local and international clinical guidelines [28,29,30]. Moreover, only a small minority of GPs agreed that vitamin B12 should be tested during check-up exams in asymptomatic patients, which seems reasonable because of the low pretest probability to detect a definitive vitamin B12 deficiency in such patients and the questioned relevance of subclinical vitamin B12 deficiency as identified by screening [6,27,31].

The usefulness of low-risk medical interventions such as vitamin prescriptions depends on costs and benefits. Direct pharmaceutical costs of vitamin B12 prescriptions are low but the largest part of expenditures caused by vitamin B12 prescriptions most likely originates from repeated intramuscular injections, which require the attention of trained healthcare professionals. In this study, we were most interested in GPs’ mindsets regarding vitamin B12 prescriptions in patients with test results within the borderline area. Since test results in the borderline area cannot rule out vitamin B12 deficiency and since anemia and peripheral polyneuropathy are clinical manifestations of vitamin B12 deficiency, treatment in these cases is undebatable [32]. However, the role of vitamin B12 regarding non-specific but prevalent central nervous symptoms is controversial. Case series have argued that cognitive symptoms, depression, or fatigue may be isolated manifestations of vitamin B12 deficiency [32], and observational studies have found cognitive decline and depression to be associated with decreased vitamin B12 [3,33,34,35]. Unfortunately, high quality randomized controlled trials have, so far, failed to demonstrate measurable benefits of vitamin B12 for cognitive outcomes and depression, suggesting that vitamin B12 deficiency may rarely be a cause of according symptoms even when vitamin B12 is in the borderline area [36]. For idiopathic fatigue, high-quality evidence suggesting beneficial effects of vitamin B12 prescription is very scarce [36]. Interestingly, in our study and in contrast to the available evidence, the majority of GPs agreed with prescribing vitamin B12 for cognitive symptoms and half of GPs agreed with prescribing it for idiopathic fatigue.

After suspected peripheral polyneuropathy, the second most common reason for testing and prescribing vitamin B12 was a risk for malnutrition/malabsorption. This seemingly large proportion may be fueled by the increasing popularity of vegan diets, which convey a substantial risk for inadequate vitamin B12 intake being devoid of animal source foods such as meat, fish, and dairy products, which are the most potent sources of vitamin B12 intake [37,38,39]. Our finding that most decisions to test and prescribe vitamin B12 are initiated by GPs and not by patients, argues for GPs themselves contributing to the trend of increased vitamin B12 testing rather than by patients’ demands. In addition, while vitamins are popular placebo interventions in general practice [40], our findings suggest GPs prescribe vitamin B12 genuinely expecting specific effects in the majority of cases. Regarding testing strategies, generally, most GPs use a multistep approach to determine vitamin B12 sufficiency, initiating with serum cobalamin as fist-line test followed by additional second-line tests. However, looking at testing strategies more precisely, we observed considerable heterogeneity, possibly indicating room for improvements. Regarding routes of application, GPs in our survey reported that, in almost 90% of the cases, intramuscular injections were used. However, recent evidence suggests that high-dose oral cobalamin is non inferior to intramuscular injections [41]. Since repeated intramuscular injections require appropriate material and trained personnel, cost saving may be achievable if GPs favored less resource-intensive routes of application. Furthermore, oral treatment regimens might improve adherence and might reduce the prevalence of vitamin B12 deficiency in this population.

Among the strengths of this study is the high response rate of 44.5% suggesting successful collection of representative data. Furthermore, this survey is currently relevant because vitamin testing and prescribing is increasing among Swiss GPs [12,13] and GPs’ mindsets are crucial determinants for understanding and, potentially, for regulating this trend if necessary. There are several limitations to the study. First, the study was conducted only in Switzerland and external validity to GPs from other healthcare systems is unknown. However, our results mirror the findings of similar studies from other countries, where variations and potential overuse of vitamin B12 testing strategies and prescriptions have been reported [42,43,44]. Secondly, several items in the questionnaire depended on estimations by GPs. Therefore, responses to these items may lack accuracy and suffer from social desirability bias accentuating results from this study. This, however, does not reduce the plausibility of our central findings in this study such as the difference mindsets among GPs regarding the necessity to test and prescribe in clinical situations without strong evidence (i.e., in idiopathic fatigue). Similarly, we did not elaborate on patient-sided factors influencing testing or prescribing decisions such as age. Furthermore, we reported several GP characteristics to be associated with willingness to test vitamin B12. These results, however, are explorative findings potentially useful to generate hypotheses, which still require confirmation and require cautious interpretation.

## 5. Conclusions

There are opportunities to reduce unnecessary spending related to vitamin B12 testing and prescribing, specifically in clinical situations, where evidence suggests futility such as subclinical vitamin B12 deficiency, cognitive complaints, and depressive symptoms. Moreover, testing strategies could be revised to reduce unnecessary testing and intramuscular injections could be reduced in favor of alternative and equally effective but less resource intensive application forms. Given the high prevalence of idiopathic fatigue in general practice and the popularity of vitamin B12 in this context, more evidence from randomized controlled trials is needed to better understand the value of ongoing testing and prescribing vitamin B12 in this clinical situation.

## Figures and Tables

**Figure 1 nutrients-13-02610-f001:**
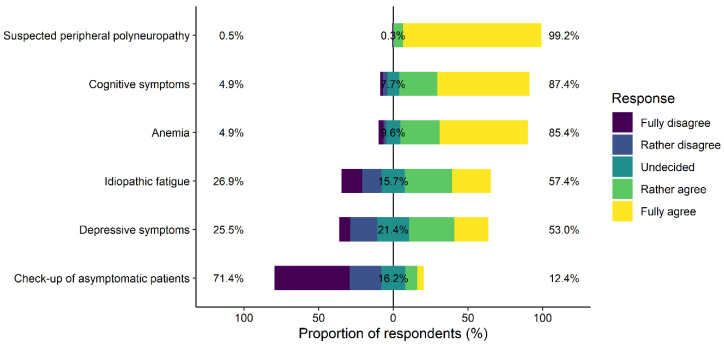
Likert plot for Question 2 (Q2): participants’ agreement to “Vitamin B12 status assessment is always indicated in this clinical situation”. The bars represent the tendency of the responses and the numbers in % at the side and in the center summarize the tendencies (left = % fully or rather disagree, center = % undecided, right = % fully or rather agree).

**Figure 2 nutrients-13-02610-f002:**
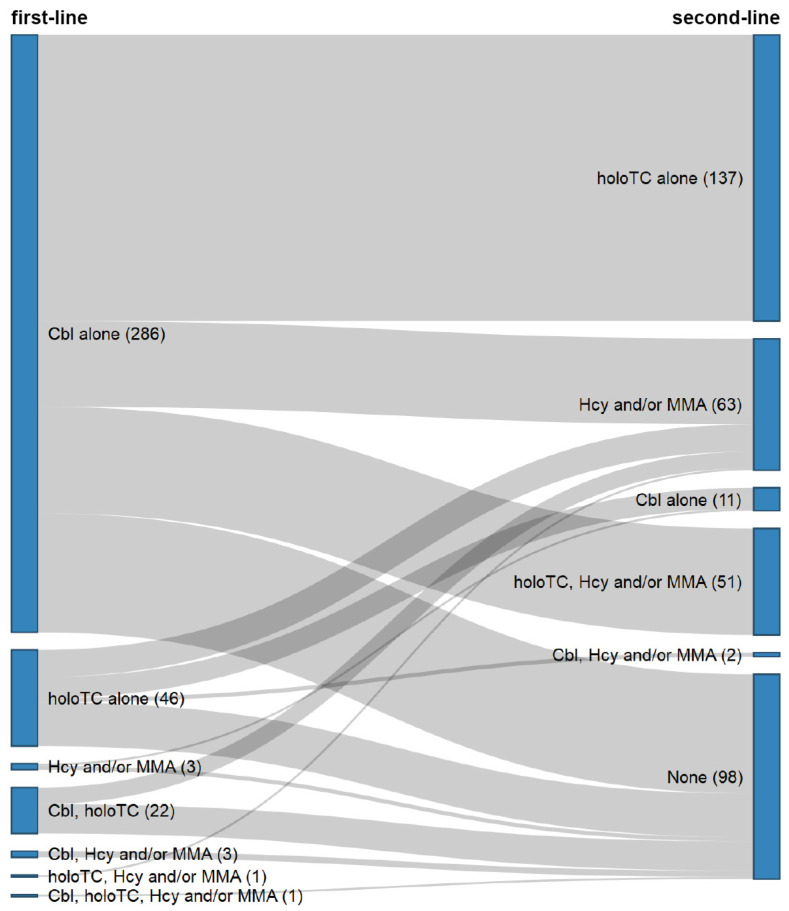
Use of serum laboratory tests for vitamin B12 status assessment. Testing strategies used for first- and second-line assessments are indicated in the left- and right-hand columns of the Sankey diagram, respectively (corresponding number of respondents in parentheses). Abbreviations: Cbl, cobalamin; Hcy/MMA, homocysteine and/or methylmalonic acid; holoTC, holotranscobalamin.

**Table 1 nutrients-13-02610-t001:** Characteristics of survey participants. Abbreviations: CAM, complementary and alternative medicine; GP, general practitioner.

Characteristic	Value
Number of respondents	376
Years of experience as a GP, median (IQR)	23.0 (16.0–30.0)
Participants that identify as female, n (%)	153 (40.7)
Practice type, n (%)	
Single	56 (14.9)
Group	306 (81.4)
Other types	2 (0.5)
Missing	12 (3.2)
Board certification in general internal medicine, n (%)	345 (91.8)
Predominantly working as GP, n (%)	341 (90.7)
Additional qualifications, n (%)	
Practices psychosomatic medicine	17 (4.5)
Practices CAM	16 (4.3)

**Table 2 nutrients-13-02610-t002:** Clinical situations appearing in the top 3 most common reasons for testing or prescribing vitamin B12 during the past year, according to GPs. For each condition, the percentage of participants indicating it as an answer to the corresponding question is reported. “Asymptomatic patients” refers to vitamin B12 testing as a screening procedure and to prescriptions after screening results suggestive for vitamin B12 deficiency.

Clinical Situation	Testing, % of Participants	Prescribing, % of Participants
Suspected peripheral polyneuropathy	77.1	75.5
Known risk for malnutrition/malabsorption	60.1	65.7
Anemia	49.7	47.1
Cognitive symptoms	34.8	35.4
Idiopathic fatigue	30.9	27.9
Depressive symptoms	11.7	10.9
Hair loss/oral aphthae	10.6	7.2
Asymptomatic patients	9.0	11.7
Other	5.3	7.7

## Data Availability

Data available on request due to restrictions, e.g., privacy or ethical. The data presented in this study are available on request from the corresponding author. The data are not publicly available due to the disclaimer of the survey.

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
