# Peer review of "Testing and Prescribing Vitamin B12 in Swiss General Practice: A Survey among Physicians"

_nutrients, 2021, doi:10.3390/nu13082610_

Round 1

Reviewer 1 Report

This paper assesses the conditions under which Swiss physicians test for vitamin B12 deficiency and prescribe vitamin B12 supplements.  The main findings were that testing and prescribing were most common for suspected peripheral neuropathy and risk of malnutrition and malabsorption, but less common for anemia, cognitive symptoms, idiopathic fatigue and depressive symptoms.  Also, serum total B12 and serum holotranscobalamin were the most often used tests of B12 status, with homocysteine and methylmalonic acid used only a small percentage of the time.  The authors conclude that these data can support educational interventions toward physicians and their practices in regard to testing for and treating B12 deficiency.   

Comments

  1. The manuscript is well written and clearly presented. I have one major question that is not addressed, and which I consider a limitation of the study. How do characteristics of the patients, in particular the age of the patients, affect the testing and prescribing attitudes of the physicians?  One would presume that an older adult (age 60 and older) might warrant more attention by the physician to issues related to B12 than younger adults.  I think this limitation should be discussed in the Discussion section.

  1. Page 3 (lines 129-130 and Table 1): I think it would be more appropriate in this day and age to refer to “participants that identify as female” as opposed to simply listing the category as “female” or “female gender”.

  1. Page 7 (lines 247-248): As written, this sentence implies that inadequate B12 intake by vegetarians and vegans is due solely to the lack of consumption of red meat. This should be expanded to include all animal source foods, including eggs, dairy, chicken, liver, and the like.

  1. Page 7 (line 254): Change “fist-line” to “first-line”.

  1. Page 7 (lines 258-259): The sentence about oral B12 being “non inferior to intramuscular injections” requires more explanation and nuance. Specifically, effectiveness of oral versus IM is dependent on the cause of the deficiency and dose amount and frequency. For patients with pernicious anemia, high oral dose B12 (1-2 mg/day) is likely as effective as monthly IM injections.  For patients with malabsorption due to atrophic gastritis or dietary deficiency, lower oral doses (25-100 micrograms per day) will likely be effective and IM injections are not necessary.

  1. Page 9: References 1 and 7 are the same reference.

Reviewer 2 Report

Bardheci et al. aimed to better understand general practitioners’ mindsets regarding vitamin B12 testing and prescribing. This paper addresses the issue where vitamin B12 is prescribed even without an unbiased measurement (that might be limited) and following complaints related to vitamin B12 deficiency. The introduction is well written, and its flow leads to the aim of this study. One limitation is the convenience sample of the practitioners, especially given the fact that they were approached via emails/online survey. Yet, about 500 practitioners are practicing medicine in different parts of Switzerland. Another limitation is the unvalidated questionnaire used for this study. A pilot test of 9 persons was performed – but only for the improvement of the survey. One striking result of this study is that the preferred route of B12 administration is by intramuscular injection (90%) and not by an oral supplement. Further comments below.

Methods:

2.1. Design, setting, and participants: The authors mentioned that ~500 practitioners from the FIRE project were approached. Please add a reference to a description of this project.

Results:

Section 3.1: can you characterize the non-responders? Did the authors have any information other than the practitioners’ email addresses?

Table 2: the table’s title (“Three most common clinical situations resulting in vitamin B12 testing and prescribing during the past year, respectively”) might be inappropriate since it describes more than 3 clinical situations.

Discussion

Line 159: In addition to these aspects of needed trained professional for the administration of intramuscular B12 and its cost, oral supplementation of vitamin B12 might also increase the adherence for the treatment and reduce the prevalence of vitamin B12 deficiency in this population.

Minor:

Please check references no. 1 and 8. These might be the same one.
